# OpenReview forum: "TraceGrad: a Framework Learning Expressive SO(3)-equivariant Non-linear Representations for Electronic-Structure Hamiltonian Prediction"
_ICML.cc/2025/Conference — ICML 2025 poster_

### Official Review · Reviewer_AGnk · 2025-02-17

**Overall Recommendation:** 4

**Summary:**

This paper presents TraceGrad, a strategy to learn SO(3)-equivariant Hamiltonian with SO(3)-invariant trace as a guiding label, built upon their mathematical relations. It aims to overcome the tradeoff between SO(3)-equivariance constraints and NNs’ nonlinear expressiveness. TraceGrad brings improvement to baseline models in ablation studies.

## update after rebuttal
The authors' answers to my questions are thorough and informative. I've maintained my score.

**Claims And Evidence:**

Claims about improvement in prediction accuracy are supported by experimental results.

**Essential References Not Discussed:**

Related works to my knowledge are comprehensively discussed.

**Experimental Designs Or Analyses:**

Experimental design and analyses look sound to me.

**Methods And Evaluation Criteria:**

The benchmark datasets are comprehensive, and evaluation criteria are reasonable.

**Other Comments Or Suggestions:**

* There’re some errors in using single quotes, e.g., Line 309.
* The discussion of equivariant operators in Sec. 5.1 seems redundant with Related Works.

**Other Strengths And Weaknesses:**

None.

**Questions For Authors:**

* The expressive is mostly added through nonlinear transformations on the invariant features. Could you comment on the inductive bias this approach assumes, and whether it’s reasonable in physics?
* Is the TraceGrad strategy applicable to learning materials properties other than full Hamiltonian?
* What’s the scope of applicability of TraceGrad? In other words, what requirements does the backbone model need to satisfy for TraceGrad to be applicable?

**Relation To Broader Scientific Literature:**

Equivariant ML models are actively developed in the AI4Materials community. I’m not sure how widely TraceGrad is applicable to such models (see Questions).

**Theoretical Claims:**

I could not check the correctness of proofs.

---

> ### Author Rebuttal · Authors · 2025-03-31
>
> 1.
> We will correct the quote errors and reduce redundancy in the discussion.
>
>
> 2.
> In this work, we explicitly introduce the trace quantity, i.e., the square of the Frobenius norm of the Hamiltonian $\mathbf{H}$, as an SO(3)-invariant quantity in our neural network. In physics, invariants often reflect the fundamental mathematical structure underlying physical laws and can serve as the foundation for deriving other quantities with equivariant properties. For example, in special relativity, the Lorentz spacetime interval remains invariant under changes in reference frames, from which one can derive equivariant quantities such as velocity and momentum. Similarly, in molecular systems, energy is an invariant quantity, and its gradient with respect to atomic coordinates yields the force, which is equivariant.
> We extend this principle—where invariant quantities induce equivariant ones—from specific physical examples to the context of neural network representation learning. This provides a structured inductive bias that guides the model in learning physically meaningful, symmetry-respecting representations.
>
> 3.
> As Reviewer gNai pointed out, the TraceGrad mechanism is highly generalizable. For other symmetry-equivariant tensorial physical quantities, the squares of their Frobenius norms are always symmetry-invariant quantities. These invariants can serve as supervision signals from which the corresponding equivariant quantities can be derived.
>
>  We validate the effectiveness of the proposed TraceGrad method on the energy/force field prediction task. Due to the limited time during the discussion period, we conducted experiments on two datasets as representative, i.e., MD17-Aspirin and MD17-Malonaldehyde [1], as a representative example. We use the same setup of this dataset as Liao and Smidt [2]. ote that in this task, the regression target $ E $ (energy) is an SO(3)-invariant quantity ($ l=0 $), while $\mathbf{F}$ (force) is an SO(3)-equivariant quantity ($ l=1 $). The model typically learns SO(3)-equivariant features $\mathbf{f}$, which are then transformed into SO(3)-invariant features, from which $E$ is regressed.
> Subsequently, the force field at a given position is obtained by differentiating $ E $ with respect to the atomic coordinates: $\mathbf{F}_i = -\frac{\partial E}{\partial \mathbf{r}_i}$, where $\mathbf{r}_i$ is the position vector of the $ i $-th atom. This approach ensures energy conservation. Given the specificity of this task, we integrate the baseline model, namely Equiformer [2] with our proposed TraceGrad method as follows:
>
> First, we use the SO(3)-equivariant features $\mathbf{f}$ encoded by the baseline model Equiformer as input, and construct SO(3)-invariant non-linear features $z$ according to our method (Sections 4 and 5 of our paper). We then use the trace quantity $\mathbf{T}$ to supervise the learning of $z$. Given $\mathbf{F}$ as a column vector with $l=1$, here $\mathbf{T}$ simplifies to $\mathbf{T} = \mathbf{F}^T \cdot \mathbf{F}$. From $z$, we induce the SO(3)-equivariant features $\mathbf{v}$ with more non-linearity, which are then fed back into the baseline model for the subsequent encoding and decoding phases, where $E$ is regressed and finally $\mathbf{F}$ is constructed from the gradients of $E$. We train Equiformer+TraceGrad under the same experimental conditions as those used in the original Equiformer paper [2], with maximum feature degree ($l_{max}$) set as 2. The experimental results are as below:
>
> For the MD17-Aspirin dataset, Equiformer achieves an energy MAE of 5.3 meV and a force MAE of 7.2 meV/Å. In comparison, Equiformer+TraceGrad  achieves lower MAE values, with 5.06 meV for energy and 5.65 meV/Å for force. For the MD17-Malonaldehyde dataset, Equiformer achieves  an energy MAE of 3.3 meV and a force MAE of 5.8 meV/Å，while Equiformer+TraceGrad, yields improved performance with an energy MAE of 3.21 meV and a force MAE of 4.68 meV/Å. **These experimental results show that our TraceGrad method improves the prediction accuracy for both energy and force, demonstrating that its effectiveness is not limited to the prediction of electronic-structure Hamiltonians and their downstream physical quantities, but has broader application potential**. Since our method is fundamentally general, we plan to extend it in the future to predict other physical properties, such as the force constant matrix, Born effective charges, and more. **We will add these experimental results and discussions in the revised paper**.
>
> [1] Chmiela et al. Towards exact molecular dynamics simulations with machine-learned force fields. Nature Communications, 2018.
>
> [2] Liao and Smidt. Equiformer: Equivariant graph attention transformer for 3d atomistic graphs. ICLR, 2023.
>
> 4.
> To apply the TraceGrad method, the backbone model only needs to be an end-to-end differentiable strictly SO(3)-equivariant neural network model, such as QHNet, DeepH-E3, or Equiformer.

---

> > ### Comment · Reviewer_AGnk · 2025-04-04
> >
> > Thanks for the explanations. I have no further comment.

---

> > > ### Author Response · Authors · 2025-04-09
> > >
> > > Dear Reviewer,
> > >
> > > We would like to sincerely thank you for your time and effort in reviewing our paper. We greatly appreciate the positive feedback and the confidence you have shown in our work . Your thoughtful comments have been incredibly helpful for this paper.
> > >
> > > Thank you again for your valuable contribution!
> > >
> > > Best regards,
> > >
> > > Authors of this paper

---

### Official Review · Reviewer_7USG · 2025-03-03

**Overall Recommendation:** 3

**Summary:**

The author propose a new technique to enhance the non-linear expressiveness for equivariant architectures. The idea is simple and straightforward, first generate an invariant feature (like energy) and equivariant feature (like position). The gradient of the invariant feature (energy) with respect to the equivariant feature (position) is an equivariant feature (force). The non-linearity is enforced on the generation of the invariant features. The author validate their methods on two K-S Hamiltonian prediction benchmarks.

Post rebuttal: The authors have addressed most of my concern.

**Claims And Evidence:**

To evaluate the effectiveness of tracegrad, more generalized tasks such as machine learning force field and quantum property regression are needed.

**Essential References Not Discussed:**

Yes, but I believe a special related work section on predicting the Hamiltonian matrix is needed.

**Experimental Designs Or Analyses:**

Yes, but an efficiency comparison is needed. I am worried that tracegrad will be siginificantly more expensive than scalar tensor interaction paradigm.

**Methods And Evaluation Criteria:**

Yes, but does regressing against  $\mathrm{tr}(H H^*)$ have any physical meaning? I think a more sensible or physical-inspired regression should be against the basis transformed version as in [1] i.e.  $\mathrm{tr}(CH H^*C^\*)$  .

[1]: Li Y, Xia Z, Huang L, et al. Enhancing the Scalability and Applicability of Kohn-Sham Hamiltonians for Molecular Systems[C]//The Thirteenth International Conference on Learning Representations.

**Other Comments Or Suggestions:**

1. Many equations are not carefully written, and things like $loss$ should be wrapped with \mathrm{} in latex.

**Other Strengths And Weaknesses:**

The idea on using the gradient as a vector feature is novel, and could serve as an useful alternative to the scalar-tensor interaction paradigm or CG-tensor products. But the application on predicting the Hamiltonian is questionable. More extensive evaluations on other tasks such as machine learning force fields, or discussion on why the method only works for the Hamiltonian is necessary.

**Questions For Authors:**

1. Can you generalize your idea, for example, the Hessian matrix? I think the Hessian matrix also has equivariant properties and could be decomposed into a set of irreducible representations. A more generalized derivation to higher-order spherical tensor could be beneficial.

2. Can you provide a detailed breakdown of your computational costs?

3. How do you approach the instability (occurrence of NaNs) of the network induced by taking the gradient? Do you make any normalizations? If so, how?

4. Can you apply your method to machine learning force field or molecular property prediction (e.g. dipole moment)?

**Relation To Broader Scientific Literature:**

It is useful for general geometric deep learning.

**Theoretical Claims:**

Why tracegrad is better than the paradigm of scalar-tensor interaction and enforcing the non-linearity on the scalar? Does the expressivity differ? Maybe a theoretical analysis could be beneficial here.

---

> ### Author Rebuttal · Authors · 2025-03-31
>
> 1.**We have found that our TraceGrad method also significantly improves energy/force prediction tasks** (please refer to the 3rd item of our response to Reviewer AGnk), demonstrating the generality of our method. We plan to follow the reviewer's suggestion and validate our method on more molecular property prediction tasks in the future work.
>
> 2.**Although both our method and that of Li et al. construct SO(3)-invariant quantities for supervised learning, the motivations and mechanisms are very different**. Li et al. construct SO(3)-invariant quantities, which are the orbital energies after diagonalizing the Hamiltonian. Using this quantity to supervise the regression model aims to ensure that the orbital energies derived from the model’s output Hamiltonian are as close as possible to the true energies.
>
> In contrast, our introduction of SO(3)-invariant quantities is intended to supervise high-quality SO(3)-invariant non-linear representations. The goal of these features is to effectively enhance non-linear expressiveness into SO(3)-equivariant features, enabling the fine regression of complex SO(3)-equivariant targets. **The work of Li et al. mentioned by the reviewer, was accepted and published online only about a week before the ICML 2025 submission deadline**, which left us with insufficient time to incorporate it into our work. However, we will certainly cite this paper and explore the possibility of combining it with our approach in future research.
>
> The SO(3)-invariant quantities we construct, $\mathbf{T} = \text{tr}(\mathbf{H} \cdot \mathbf{H}^\dagger)$, quantify the total coupling strength encoded in the Hamiltonian matrix, representing the overall amplitude of electronic interactions in the system. It is a symmetry-invariant scalar that reflects the global energy scale of $\mathbf{H}$ and serves as a physically meaningful regularization target.
> From a machine learning perspective, it serves as an excellent target to supervise SO(3)-invariant representations.
> **Moreover, this construction is easily extendable to other physical quantities**. For example, in force field prediction tasks, one could construct the trace quantity of the force, which reflects the strength of the force,
>  as a supervision signal, train high-quality SO(3)-invariant feature representations, and then induce SO(3)-equivariant features through the gradient mechanism.
>
> In some cases, using $tr( \mathbf{C} \mathbf{H} \mathbf{H}^\dagger \mathbf{C}^\dagger)$ can also be a reasonable choice. In particular, when $\mathbf{C}^\dagger \mathbf{C} = \mathbf{I}$, this expression reduces exactly to our definition, $tr( \mathbf{H} \mathbf{H}^\dagger)$. However, its applicability is not as general as that of our proposed formulation, and it is also less straightforward to implement in practice. For instance, in the case of other tensorial physical quantities, such as the force $\boldsymbol{F}$, which is a vector, or the electron-phonon coupling tensor, which is a rank-3 tensor, our definition remains valid, whereas $tr( \mathbf{C} \mathbf{H} \mathbf{H}^\dagger \mathbf{C}^\dagger)$ may no longer be applicable. Therefore, our trace-based construction offers a unified, symmetry-invariant, and easily implementable approach for supervising or regularizing such tensor quantities across a wide range of physical systems.
>
> 3.**A theoretical analysis on the advantages of the TraceGrad method**.
> First, our method constructs SO(3)-invariant quantities, i.e., the trace quantity, to directly supervise SO(3)-invariant features, allowing for the effective learning of informative SO(3)-invariant features that capture the intrinsic symmetry properties of the mathematical structure of $\mathbf{H}$. This helps the model learn informative invariant features and ultimately deliver them to the equivariant features to assist in Hamiltonian prediction. Second, the proposed gradient mechanism, i.e., $\mathbf{v} = \frac{\partial z}{\partial \mathbf{f}}$, where $\mathbf{v}$ and $\mathbf{f}$ are used to regress $\mathbf{H}$ and $z$ is used to regress $\mathbf{T}$, reflects the partial derivative relationship between $\mathbf{H}$ and $\mathbf{T}$, i.e., $\mathbf{H} = \frac{\partial \mathbf{T}}{\partial Conj(\mathbf{H})}$, where $Conj(\cdot)$ denotes the complex conjugate, imposing stronger physical constraints on the relationships between the components of the equivariant features. In contrast to the conventional gated activation mechanism, which can be expressed as $\mathbf{v} = z \cdot \mathbf{f}$, this approach enables effective joint learning of $z$ and $\mathbf{v}$, with supervision provided by $\mathbf{T}$ and $\mathbf{H}$.
>
> 4.We will present the related works on Hamiltonian prediction in a dedicated section for clarity.
>
> 5.We will revise the equation rendering to ensure standard formatting.
>
> 6.Please refer to the 1st item of responses to Reviewer gNai for the discussion on computational cost.
>
> 7.We use layer normalization techniques to stabilize the training process.

---

> > ### Comment · Reviewer_7USG · 2025-04-07
> >
> > I thank the authors for their comment. I will increase my score to 3.

---

> > > ### Author Response · Authors · 2025-04-09
> > >
> > > Dear Reviewer,
> > >
> > > We would like to sincerely thank you for your thoughtful feedback and the constructive comments you have provided. Your suggestions have been immensely helpful, and we truly appreciate the time and effort you dedicated to reviewing our paper. We are grateful for the higher rating and the confidence you have shown in our work.
> > >
> > > Best regards,
> > >
> > > Authors of this paper

---

### Official Review · Reviewer_gNai · 2025-03-14

**Overall Recommendation:** 3

**Summary:**

This paper introduces TraceGrad, a framework that integrates strong non-linear expressiveness with strict SO(3)-equivariance for electronic structure Hamiltonian prediction. The approach first constructs theoretical SO(3)-invariant trace quantities derived from Hamiltonian targets, using them as supervisory signals to learn invariant features. A gradient-based mechanism is then employed to generate SO(3)-equivariant encodings of varying degrees from these learned invariant features. Empirical evaluations on eight benchmark datasets demonstrate improvements in predicting physical quantities and accelerating density functional theory (DFT) computations.

**Claims And Evidence:**

The paper makes two key claims:
* Addressing the Challenge of Combining Non-Linear Expressiveness with SO(3)-Equivariance
    * The authors propose a novel approach to this challenge by systematically bridging SO(3)-invariant and SO(3)-equivariant representations.
    * The framework first supervises SO(3)-invariant features to ensure strong non-linear expressiveness and subsequently derives SO(3)-equivariant representations through a gradient-based mechanism.
    * The mathematical derivations in Section 4 provide a solid theoretical foundation, and empirical results in Section 6 and Appendix H support the claim.
* Significant Performance Gains in Hamiltonian Prediction on Eight Benchmark Datasets
    * The authors report that TraceGrad outperforms state-of-the-art methods across eight datasets from the DeepH and QH9 benchmark series.
    * This claim is supported by empirical results in Section 6 and Appendix H, which demonstrate improved prediction accuracy and DFT acceleration.

**Essential References Not Discussed:**

No essential references appear to be missing from the discussion, as the submission adequately contextualizes its contributions.

**Experimental Designs Or Analyses:**

* The experimental setup is well-structured and comprehensive.
* However, there is no detailed analysis of parameter counts, which is crucial for assessing model expressiveness. The authors state that  $g_{\text{nonlin}}(\cdot)$  is implemented as a three-layer fully connected module with a large hidden size, injecting significantly more parameters into the architecture. Since larger models tend to have higher expressiveness, a fair comparison requires increasing the parameter counts of baseline models (e.g., by widening hidden layers or deepening architectures). Providing such comparisons would clarify TraceGrad’s advantages over other methods.

**Methods And Evaluation Criteria:**

* Methods: The approach is well-grounded in equivariant neural networks for Hamiltonian prediction.
* Evaluation Criteria: The chosen metrics align with standard benchmarks in quantum chemistry.

**Other Comments Or Suggestions:**

* A detailed parameter count analysis for various model variants would strengthen the paper and clarify the technical contribution of TraceGrad.

**Other Strengths And Weaknesses:**

Strengths
* The paper is well-organized and clearly written.
* The proposed method is theoretically sound and provides an innovative solution to the equivariance-expressiveness tradeoff.

**Questions For Authors:**

* Have the authors analyzed the impact of the hidden size of  $g_{\text{nonlin}}(\cdot)$ on model performance?

**Relation To Broader Scientific Literature:**

TraceGrad’s contribution to equivariant graph neural networks is general and could inspire new model designs for other molecular property prediction tasks beyond Hamiltonian prediction.

**Theoretical Claims:**

* The construction of SO(3)-invariant trace quantities and the gradient-based mechanism linking invariant and equivariant representations are mathematically sound.
* The authors assert in Introduction and Remark 4.3 that the gradient mechanism induces expressive SO(3)-equivariant representations while maintaining physical interpretability, providing an advantage over gated mechanisms. Empirical ablation results (Appendix H) partially support this claim, but it remains unclear whether model variants using gating mechanisms have comparable parameter counts to those using TraceGrad. A direct comparison in terms of model complexity would strengthen this argument.

---

> ### Author Rebuttal · Authors · 2025-03-31
>
> 1.**Response to the reviewer's question about the computational burden**:
> First, the branch decoding the trace quantity $\mathbf{T}$ from the SO(3)-invariant features $z$ is only required during the training phase and does not need to be activated during inference; thus, the parameters associated with this branch are unnecessary at inference time. Second, the key difference between our proposed gradient-based mechanism for constructing non-linear SO(3)-equivariant features and the original gated mechanism is that the feature construction has changed from $\mathbf{v} = z \cdot \mathbf{f}$ to $\mathbf{v} = \frac{\partial z}{\partial \mathbf{f}}$. Given the network that learns the SO(3)-invariant features $z$, the gradient operation itself does not introduce additional parameters.
>
> The reviewer's suggestion to analyze the computational cost is reasonable. Unfortunately, existing automated FLOPs counting tools like fvcore, torchprofile, and torchstat do not support precise quantification of the computational complexity for frameworks based on equivariant neural network packages such as E3NN. Therefore, in our paper, we measure the computational efficiency of our method by testing the average inference time under the same GPU/CPU hardware conditions.
>
> In the paper’s Appendix J,  we provide a comprehensive comparison of GPU computational costs and corresponding accuracy for different models across four representative databases. From the experimental results in Table 7 of our paper, **we find that incorporating our TraceGrad method results in only a slight increase in inference time compared to the baseline models, i.e., DeepH-E3 or QHNet**. Given the substantial accuracy improvements introduced by the TraceGrad method, this minor increase in computational time is considered acceptable for practical applications. **In contrast, simply doubling the depth of DeepH-E3 or QHNet leads to a significant rise in inference time while providing only limited accuracy improvements.**
> In contrast, DeepH-E3+TraceGrad and QHNet+TraceGrad exhibit significantly better accuracy performance compared to DeepH-E3$^{\times 2}$ and QHNet$^{\times 2}$, respectively. At the same time, the inference times of DeepH-E3+TraceGrad and QHNet+TraceGrad are considerably lower than those of DeepH-E3$^{\times 2}$ and QHNet$^{\times 2}$, respectively. **These findings highlight the superiority of the TraceGrad method in enhancing model expressiveness and improving accuracy performance while maintaining computational efficiency.**
>
> To further address the reviewer's concern regarding the efficiency differences between the classical gated mechanism and our TraceGrad method, we introduce an additional experimental setup to evaluate inference efficiency. Specifically, we test QHNet+Gate, where the gradient-based mechanism for constructing the non-linear equivariant feature $ \mathbf{v} $ is replaced with a classical gated mechanism, following the definition provided in Appendix H of the paper.
> Experimental results show that **QHNet+Gate** achieves inference times of 0.243s and 0.184s on the QS and QD datasets, with $ \text{MAE}^H_{\text{all}} $ values of 1.796 meV and 4.217 meV, respectively. In comparison, **QHNet+TraceGrad** takes 0.248s and 0.187s on the same datasets, with $ \text{MAE}^H_{\text{all}} $ values of 1.191 meV and 2.819 meV, respectively. These results demonstrate that the TraceGrad method introduces only a minor increase in inference time compared to the traditional gated mechanism, while achieving significant improvements in accuracy.
>
> In addition to reporting the GPU inference time, **we also present the inference times of QHNet and QHNet+TraceGrad on a single CPU thread in the paper’s Appendix K**. Experimental results show that while combining TraceGrad introduces only a slight increase from the inference time of QHNet on the CPU, it delivers significant improvements in accelerating the convergence of DFT methods. **Notably, the time saved by TraceGrad for DFT calculations far exceeds the minimal additional time introduced by TraceGrad for the deep model's inference.**
>
> 2.We agree with the reviewer’s view that TraceGrad’s contribution to equivariant graph neural networks is indeed general. In addition, **we have found that it demonstrates its effectiveness on another task, namely energy and force field prediction**. For more details, please refer to the 3rd item of our response to Reviewer AGnk .
>
> 3.We have reduced the hidden size of $g_{\text{nonlin}}(\cdot)$ by half (from 1024 to 512) and conducted experiments on the QHNet-Stable (QS) database. The experimental results show that the $MAE^H_{\text{all}}$ metric increase from 1.191 to 1.347, but still significantly outperforms the baseline method (1.962). This suggests that while the parameter size does impact the accuracy improvements introduced by our method, the effect is relatively limited.

---

> > ### Comment · Reviewer_gNai · 2025-04-03
> >
> > Thanks for the authors' response. Some of my concerns have been addressed.
> > * Regarding R1 and R3, the response partially resolves my concerns. While the ‘SO(3)-invariant decoder’ is not activated during inference, the branch that generates SO(3)-invariant features z still introduces additional parameters. Given that model expressiveness is closely tied to the total number of trainable parameters, disregarding these so-called extra parameters when discussing expressiveness and model complexity is not entirely appropriate. A direct comparison of parameter counts should be provided to clarify this point.
> > * Regarding R2, I appreciate the authors’ efforts in exploring the model’s application to other tasks.
> >
> > Accordingly, I will maintain my rating.

---

> > > ### Author Response · Authors · 2025-04-09
> > >
> > > Dear Reviewer,
> > >
> > > We would like to express our sincere gratitude for your time and effort in reviewing our paper. We truly appreciate your constructive feedback and the positive comments you provided.
> > >
> > > Best regards,
> > >
> > > Authors of this paper

---

### Official Review · Reviewer_1Tq8 · 2025-03-19

**Overall Recommendation:** 3

**Summary:**

This paper proposes to enhance the Hamiltonian prediction networks with additional invariant supervision and an additional gradient branch. The authors observe that the trace of $H H^T$ is rotation invariant and can be used to supervise the learning of zero order features. Additionally, the gradient of a network using the zero order feature as input w.r.t. the zero and non zero order features is rotation equivariant, so it can be added back to the original equivariant feature. The authors combine these two techniques and test with DeepH-E3 and QHNet models and test on material datasets and small molecule datasets.

**Claims And Evidence:**

- Using the trace of $H H^T$ to supervise zero order feature learning seems to be a reasonable design.

- However, I have a doubt regarding the gradient part. Wouldn't taking the local gradient w.r.t. to the equivariant feature result in a linear scaling of the original equivariant feature? i.e., it won't change the direction of the equivariant feature. For example, for order 1 feature $v\in \mathbb{R}^3$, the zero-order feature from the CG decomposition of $v\bigotimes v$ would be proportional to the dot product $v\cdot v$. As a result, if we take the gradient w.r.t. $v$, the quadratic term in dot product will reduce to be linear w.r.t. $v$. This seems to hold even after applying the non-linear neural networks due to the chain rule. So the final equivariant feature from the gradient branch would be a scaling of the original equivariant feature.

**Essential References Not Discussed:**

- Not I am aware of.

**Experimental Designs Or Analyses:**

- The experimental design seems to be sound.

**Methods And Evaluation Criteria:**

- The benchmarked dataset includes both periodic and non-periodic systems, which is a strength.

**Other Comments Or Suggestions:**

- I did not notice typo.

- Some ablation studies like (+Trace, +Gate, +Grad)  might be presented in the main text.

**Other Strengths And Weaknesses:**

- The experimental results are promising.

- The theoretical analysis of the proposed gradient branch might be enhanced.

**Questions For Authors:**

- I noticed the ablation studies in the appendix (Table 5) about DeepH-E3 (+Trace, +Gate, +Grad), which I think are important results. Did the authors observe similar trends for QH9 datasets?

**Relation To Broader Scientific Literature:**

- The related works are adequately discussed.

**Theoretical Claims:**

- Most equations are descriptive rather than proving something.

---

> ### Author Rebuttal · Authors · 2025-04-01
>
> 1.**Clarification regarding whether the gradient mechanism can change the direction of features**: In Theorem 2 of our paper, for simplicity  and to highlight the core ideas, we selected $\textbf{f}$ as a basic feature component of degree $l$. In this case, applying the gradient operation directly to $\textbf{f}$ results in $\textbf{v}$ that only changes its magnitude but not its direction. However, in the actual application of this theory, a series of feature components $\textbf{f}_1, \textbf{f}_2, \cdots, \textbf{f}_C$ with degree $l$ are applied with the gradient operation to obtain corresponding $\textbf{v}_1, \textbf{v}_2, \cdots, \textbf{v}_C$, a new series of feature components.
>
> These components are then combined to form a new feature, $\textbf{v} = \sum_{1 \leq c \leq C} w_c \textbf{v}_c$, where $w_c$ represents the combinational coefficients. In this case, both the direction and magnitude of the new feature $\textbf{v}$ are different from those of the original features $\textbf{f}_1$, $\textbf{f}_2$, ..., $\textbf{f}_C$. In fact, our method involves a more in-depth extension (please refer to lines 178-249 in section 5.1 of our paper for details). For different feature components, e.g., $\textbf{f}^{(k)l_i}$ and $\textbf{f}^{(k)l_j}$ in Eq. (2) of our paper, as long as they share the same degree $(l_i=l_j)$, we can construct SO(3)-invariant and SO(3)-equivariant features using them, even if their magnitudes and directions are different. Through such operations and  linear combinations, we can ultimately encode new features with flexible direction and magnitude, significantly enhancing the expressive power of the features. **Therefore, while the gradient operation on a single feature component itself does not change its direction, by combining multiple feature components and leveraging gradients and linear combinations, we can encode richer, more expressive feature representations with varying directions and magnitudes.**
>
> We would like to thank the reviewer for their valuable feedback. We will clarify this analysis in the theoretical section of our paper to ensure a more accurate and comprehensive understanding of the gradient-based mechanism we proposed.
>
> 2.We will move the ablation studies into the main text of the revised version of our paper.
>
> 3.We have added ablation studies on the QH9-Stable (QS) database, where the notation for each experimental setup is consistent with the definitions provided in Appendix H. The results are summarized in the table below. As can be observed,  the key conclusion on this database is consistent with that in the DeepH-E3 benchmark series: each individual component (Trace and Grad) of our method contributes positively to the overall performance, and their combination leads to further improvements. Notably, the gradient-based mechanism (Grad) consistently achieves higher accuracy compared to the gated mechanism (Gate).
>
> **Table**: Experimental results measured by the $MAE^H_{all}$, $MAE^H_{diag}$, $MAE^H_{nondiag}$, $MAE^{\epsilon}$, and $Sim(\psi)$ metrics on the QH9-stable (QS)  database using 'ood'  split strategy. $\downarrow$ means lower values correspond to better accuracy, while $\uparrow$ means higher values correspond to better performance. The units of MAE metrics are meV, while $Sim(\psi)$ is the cosine similarity which is dimensionless.
>
> | **Method**           | $MAE^H_{all}$ ↓ | $MAE^H_{diag}$ ↓ | $MAE^H_{nondiag}$ ↓ | $MAE^{\epsilon}$ ↓ | $Sim(\psi)$ ↑ |
> |----------------------|--------------------|----------------------|----------------------------|----------------|------------|
> | QHNet (Baseline)     | 1.962              | 3.040                | 1.902                      | 17.528         | 0.937      |
> | QHNet+Trace          | 1.874              | 2.936                | 1.815                      | 16.724         | 0.940      |
> | QHNet+Gate           | 1.796              | 2.845                | 1.741                      | 15.696         | 0.940      |
> | QHNet+Grad           | 1.604              | 2.587                | 1.558                      | 11.393         | 0.942      |
> | QHNet+TraceGate      | 1.516              | 2.426                | 1.465                      | 10.568         | 0.945      |
> | QHNet+TraceGrad      | **1.191**          | **2.125**            | **1.139**                  | **8.579**      | **0.948**  |

---

### Decision · Program_Chairs · 2025-05-01

**Decision:**

Accept (poster)

**Comment:**

The paper presents a mean-field electronic Hamiltonian prediction network that guarantees the equivariance of the high-order tensors. The equivariance is achieved by taking the gradient of an invariant feature, similar to the vector-equivariant case. The invariant feature can be processed by arbitrary nonlinearity to provide expressiveness, while also holds the interaction results between tensors of different orders through tensor product. In addition, the final resulting invariant feature is guided by the trace of the Hamiltonian, an invariant quantity. The design makes a good sense, and experiments on common benchmarks show improved accuracy than existing models.

Reviewers asked for more details in more aspects on assessing the work.
* Training and inference efficiency, as asked by Reviewers gNai and 7USG. This is a reasonable ask as gradient-based models typically require more computation and memory. The authors provided a few running time comparisons, showing that not significantly more time cost is induced. Nevertheless, to make it solid, the authors need to further append hardware details right after these results.
* Understanding the expressiveness of the model where tensorial feature is updated with gradient w.r.t a scalar (Reviewers 1Tq8, 7USG). The reply regarding whether the gradient expression can rotate the input tensor makes sense to me, while using $\\mathbf{H} = \\frac{\\partial\\mathbf{T}}{\\partial \\mathrm{Conj}(\\mathbf{H})}$ to explain the mutual benefit between $\\mathbf{H}$ and $\\mathbf{T}$ supervision seems a bit stretching, as the gradient in the model is not directly taken on $\\mathbf{T}$ and $\\mathbf{H}$ is not directly produced from the gradient. I hope the authors could provide further comment on the expressiveness/universality, following the consideration that the gradient w.r.t a vector can only model a conservative vector field, and if the combination of gradient tensors can circumvent a similar constraint. (This may be out of the scope, though.)
* Reviewer gNai asked about the model size (parameter count) in empirical comparisons. While the authors claimed no additional parameters are introduced when implementing the TraceGrad operation, I agree with the reviewer that there are still some more. I hope the authors could provide further information in the final paper.
* Energy/force prediction performance as asked by Reviewer 7USG. The provided results during rebuttal seem supportive.
* Literals in equations should be put upright.

Overall, the issues seem minor compared with the contribution.